# SOTERIA: IN SEARCH OF EFFICIENT NEURAL NETWORKS FOR PRIVATE INFERENCE

## ABSTRACT

Consider ML-as-a-service where a cloud server hosts a trained model and offers prediction (inference) service to users. In this setting, our objective is to protect the **confidentiality** of both the users' input queries as well as the model parameters at the server, with modest computation and communication overhead. Prior solutions primarily propose fine-tuning secure multi-arty computation protocols to make them efficient for known *fixed* model architectures. However, the model itself is not originally *designed* to operate efficiently with existing efficient cryptographic computations. We observe that the network architecture, internal functions, and parameters of a model, which are all chosen during training, significantly influence the computation and communication overhead of a cryptographic method, during inference. We propose SOTERIA — a training method to construct model architectures that are by-design efficient for private inference. We use neural **architecture search** algorithms with the dual objective of optimizing the accuracy of the model and the overhead of using cryptographic primitives for secure inference. We select garbled circuits as our underlying cryptographic primitive, and empirically evaluate SOTERIA to compare it with the prior work. Our results confirm that SOTERIA is effective in balancing performance and accuracy.

## 1 INTRODUCTION

Machine learning models are susceptible to several security and privacy attacks throughout their training and inference pipelines. Defending each of these threats require different types of security mechanisms. One important requirement is that the sensitive input data as well as the trained model parameters remains confidential during the inference time. In this paper, we focus on private computation of inference over deep neural networks, which is the setting of machine learning-as-a-service. Consider a server that provides a machine learning service (e.g., classification), and a client who needs to use the service for an inference on her data record. The server is not willing to share the proprietary machine learning model, underpinning the service, with any client. The clients are also unwilling to share their sensitive private data with the server. In addition, we assume the two parties do not trust, nor include, any third entity in the protocol. In this setting, our first objective is to design a secure protocol that protects the confidentiality of client data as well as the prediction results against the server who runs the computation. The second objective is to preserve the confidentiality of the model parameters with respect to the client. These can be achieved using secure multi-party computations. We consider an honest-but-curious threat model.

A number of techniques provide data confidentiality while computing and thereby allow *private computation*. The techniques include computation on trusted processors such as Intel SGX Hunt et al. (2018a); Ohrimenko et al. (2016), and computation on encrypted data, using homomorphic encryption, garbled circuits, secret sharing, and hybrid cryptographic approaches that jointly optimize the efficiency of private inference on neural networks Yao (1986); Gentry (2009b); Yao (1982); Brakerski et al. (2011); Beaver (1992). To provide private inference with minimal performance overhead and accuracy loss, the dominant line of research involves adapting cryptographic functions to (an approximation of) a given *fixed* model Liu et al. (2017); Mohassel & Zhang (2017); Mishra et al. (2020); Rouhani et al. (2018); Riazi et al. (2019); Chandran et al. (2019); Juvekar et al. (2018); Riazi et al. (2018). However, the alternative approach of *searching or designing* a network architecture for a given set of efficient and known cryptographic primitives has not received much attention Ghodsi et al. (2020).

Note that secure multi-party computations protect the confidentiality of the inputs to the protocol, and its intermediate computations, and cannot prevent *inference* attacks that exploit the output of the protocol to infer information about its inputs. We thus emphasize that the protection against the indirect inference attacks that aim at reconstructing model parameters Tramèr et al. (2016) or its training data Shokri et al. (2017), by exploiting model predictions, is not our goal and out of scope for this work.

**Our Contributions.** We approach the problem of private inference from a novel perspective where instead of modifying cryptographic schemes to support neural network computations, we advocate modification of the training algorithms for efficient cryptographic primitives. In a nutshell, we advocate co-designing and co-optimizing models and cryptographic operations for efficient private inference. Research has shown that training algorithms for deep learning are inherently flexible with respect to their neural network architecture. This means that different network configurations can achieve similar level of prediction accuracy. We exploit this fact about deep learning algorithms and investigate the problem of optimizing deep learning algorithms to ensure efficient private computation.

To this end, we present SOTERIA — an approach for constructing deep neural networks optimized for performance, accuracy and confidentiality. Although SOTERIA could leverage any of the available cryptographic primitives or their combination, we select garbled circuits as its main building block to address the *confidentiality* concern. Garbled circuits (GC) are known to be efficient and allow generation of constant depth circuits even for non-linear functions. We show that neural network algorithms can be optimized to efficiently execute garbled circuits while achieving high accuracy guarantees. We observe that the efficiency of evaluating an inference circuit depends on two key factors: the model parameters and the network structure. With this observation, we design a regularized architecture search algorithm to construct neural networks. SOTERIA selects optimal parameter sparsity and network structure with the objective to guarantee a balance between *performance* and model *accuracy*. We summarize our contributions below:

- We propose a neural architecture search based approach called SOTERIA for designing models that guarantee a balance between prediction utility and computation cost while maintaining confidentiality of the model parameters and the input during inference.

- We build SOTERIA models using garbled circuits for both ternary neural networks as well as the model architecture identified using network architecture search approach.

- We evaluate SOTERIA on the benchmark datasets used by all the prior work, and observe that our approach provides the flexibility to tune utility and efficiency parameters based on the requirement of the underlying scenario. Our complete anonymous source code is available on github.[1]

The two fields of research on neural architecture search Liu et al. (2019); Zoph & Le (2016); Elsken et al. (2018) and secure multi-party computation Yao (1986); Gentry (2009b); Yao (1982); Brakerski et al. (2011); Beaver (1992) are rapidly growing. SOTERIA enables us to take advantage of the advancements in these fields to systematically design more efficient algorithms for complex machine learning tasks.

## 2 SOTERIA

We design SOTERIA to automatically learn the model architecture and its connections so as to optimize the cost of private inference in addition to optimizing accuracy. This approach is different than simply fine-tunning or compressing a model, as we aim to include the cost of private computation as part of the objective of *architecture learning* and *parameter learning* of the model. SOTERIA is built on top of two well-established classes of machine learning algorithms: neural architecture search algorithms, and ternary neural network algorithms.

**Neural architecture search for efficient private inference.** Architecture search algorithms for neural networks are designed to replace the manual design of complex deep models. The objective is to learn a model structure that gives high accuracy when trained on the training set Elsken et al.

---

[1]Source code of SOTERIA: `https://github.com/SoteriaAnonymous/Soteria`

(2018); Zoph & Le (2016); Pham et al. (2018); Liu et al. (2019). They automatically construct the model architecture by stacking a number of *cells*. Each cell is a directed acyclic graph, where each node is a neural *operation* (e.g., convolution with different dimensions, maxpool, identity). During the search algorithm, we use stochastic gradient descent to continuously update the scores associated with different candidate operations for each connection in the internal graph of a cell, as to maximize the accuracy of the model on some validation dataset.

In SOTERIA, we regularize the computation of the connection scores over candidate operations, with factor $\lambda$. We penalize each operation proportional to its computation and communication overhead when garbled. Larger values of $\lambda$ result in models that prefer efficiency of private inference over accuracy. As we balance the trade-off between accuracy and performance, SOTERIA can construct models which *by design* satisfy the requirements of our system.

**Ternary (Sparse Binary) Neural Network.** To build a system that enables *efficient* private inference, we aim to reduce the number of parameters contained in the model. One approach to accomplish this is to first train the model, and then compress it afterwards. However, this method might not result in the highest accuracy that could be achieved if we were to construct the model for this purpose from the beginning. To better align with our approach of *constructing* model architectures, we aim to learn model structures which are accurate, sparse, and are efficient when implemented as garbled circuits. More specifically, in SOTERIA, we train models with ternary parameters $(-1, 0, +1)$ Li et al. (2016) and binary activation functions. This allows us to leverage the benefits for BNNs, as discussed in Section B, with the benefit of a potentially smaller circuit (due to the model's sparsity). We incorporate Ternary Neural Networks, or TNNs, into our regularized architecture search algorithm to find cells containing only ternary convolution and max-pooling layers that operate on binary inputs and ternary parameters.

## 3 EMPIRICAL EVALUATION

In this section, we evaluate the efficiency of our method in two main ways. First, we show how using ternary neural networks on fixed model architectures, as used in the prior work Riazi et al. (2019); Mohassel & Zhang (2017); Rouhani et al. (2018); Juvekar et al. (2018), can reduce the overhead of secure inference on sparse neural networks. Second, we present the performance of SOTERIA architectures, in which model complexity is optimized along with the model accuracy.

**Implementation.** We first build a representation of the model and parameters in SystemVerilog and then convert the model into a digital circuit supported by TinyGarble Songhori et al. (2015), and implementation of the Garbled Circuit (GC) protocol. We synthesize and optimize our circuit (using Synopsys Design Compiler) to use circuit elements supported by TinyGarble. In the first step, we design a collection of parameterized components (notably dot product, and maxpool) to use as building blocks in our architecture search algorithm. Each component is flexible and can efficiently accept arbitrary size input and output, and is composed to form the complete model. In a general setting, hardware-level code is typically straight-forward to generate. However, to enable TNNs with SOTERIA, we have to dynamically define the sparsity of the modules depending on the result of model training and architecture search. Along with the parameter data, we define the sparsity information which is used during `generate` phases in SystemVerilog to build the sparse network in hardware (taking advantage of the 0-valued parameters). Altogether, this allows us to build and evaluate SOTERIA models.

### 3.1 EXPERIMENTAL SETUP

We need to evaluate SOTERIA on datasets which are used in the prior work to be able to perform a comprehensive comparison between the methods (see Table 3 for all the configurations). We evaluate our work on MNIST and CIFAR10 image classification datasets, as they are the only benchmark datasets which are extensively used in the literature to evaluate the performance of cryptographically secure neural network schemes. We run our experiments on an AWS c5.2xlarge instance, running Ubuntu 18.04 LTS on an Intel Xeon 8124M at 3.0 GHz. We use PyTorch 1.3 Pyt, a deep learning framework to implement our architecture search algorithm and train the ternary models. We use Synopsys Design Compiler Syn, version L-2016.03-SP5-2, to synthesize SystemVerilog code into

the gate-level netlist. Our synthesis runs the TinyGarble gate library infrastructure[2]. We compute the number of non-XOR gates in the generated boolean circuit netlist as a measure of its complexity. We measure the exact performance of SOTERIA as its runtime during the offline and online phases of the protocol, and its communication cost. We present the experimental setup of prior work and SOTERIA, including their CPU specification and link to available software codes, in Table 7. We also present the details of all neural network architectures which we evaluate in this paper in Table 6.

## 3.2 RESULTS FOR GC ON TERNARY NEURAL NETWORKS

In this subsection, we illustrate the benefits of taking advantage of the inherent sparsity (fraction of parameters with weight 0) found in ternary models, and analyze the effect of the scale of the network in the tradeoff between model accuracy and performance of private inference.

**Sparsity.** The sparsity of the trained model allows for the reduction in the amount of computation needed in the resulting privacy-preserving Garbled Circuit implementation. To demonstrate this, Figure 1a lists the number of non-XOR gates needed when building a collection of ternary neural networks with a 4-kernel $3 \times 3$ convolution operation, an input of size $32 \times 32 \times 3$, and a padding of size 1. We randomly set a fraction of the parameters to zero to manually control the sparsity of the model. A BNN model is equivalent to the case where the sparsity is 0. With increasing sparsity, we see a direct benefit in the reduction of non-XOR gates (reducing those gates that require expensive computation in a GC implementation) in the resulting computation. This can result in reducing both the communication overhead and the inference runtime, which we see as we train large models used in SOTERIA throughout the rest of this section.

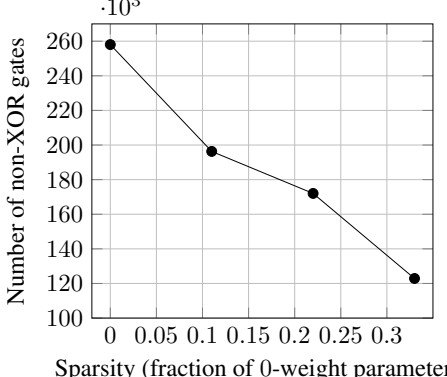
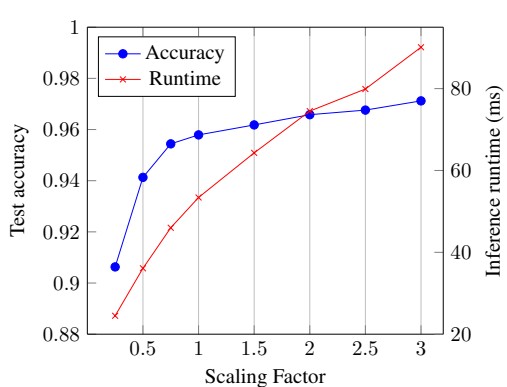

(a) Sparsity of a model versus its circuit complexity.

(b) Test accuracy versus inference runtime.

Figure 1: (a) We measure sparsity as the fraction of model parameters with 0 weight. We quantify circuit complexity as the number of non-XOR gates. The numbers are computed on a ternary neural network with a $3 \times 3$ convolution operation with 4 kernels and a $32 \times 32 \times 3$ input. For this experiment, we assign 0 weights to a random set of parameters, to get different levels of sparsity and corresponding number of non-XOR gates. (b) Test accuracy versus inference runtime for a ternary neural network trained on a fixed MNIST (m1) architecture, in various scale.

Note that while training a TNN, we cannot control the sparsity of the network. In Table 1, we show the result of training both binary and ternary models on MNIST dataset, model architecture m3 (see Table 6 for the model details). The GC costs of the components of the network for both BNNs and TNNs are listed, with the trained ternary model showing significant sparsity (almost 0.3). This level of sparsity results in fewer circuits for the trained model, and a lower overall inference cost for the GC protocol when using these ternary neural networks. This is reflected in the smaller number of non-XOR gates in the ternary circuits needed for the convolution and fully connected operations. The costs for the maxpool operation will remain the same, as it does not contain any learnable parameters.

---

[2]We base our work on TinyGarble with gitid 21ecca7b75b33fd7508771fd35f03657dd44e5e from `https://github.com/esonghori/TinyGarble`.

Table 1: Performance of inference on garbled circuit models with binary versus ternary parameters. We show the independent offline and online runtimes, communication costs and number of non-XOR gates for three different types of operations. The operations are taken from MNIST (m3) network, trained in both binary (BNN) and ternary (TNN) configurations using a scaling factor of 1.

| Operation Type | Size | Runtime (ms) | | | | | | Communication (KB) | | Number of non-XOR gates | | TNN Sparsity |
|---|---|---|---|---|---|---|---|---|---|---|---|---|
| | | Offline | | Online | | Total | | | | | | |
| | | BNN | TNN | BNN | TNN | BNN | TNN | BNN | TNN | BNN | TNN | |
| Convolution | Input: $12 \times 12 \times 16$ Padding: 0 Window: $5 \times 5$ Kernels: 16 | 39.16 | 28.27 | 52.87 | 38.17 | 92.03 | 66.43 | 2,572 | 1,876 | 589,824 | 425,558 | 0.27 |
| Maxpool | Input: $8 \times 8 \times 16$ Window: $2 \times 2$ | 0.35 | | 0.57 | | 0.92 | | 34 | | 768 | | N.A. |
| Fully Connected | Input: 100 Nodes: 10 | 0.49 | 0.34 | 0.83 | 0.60 | 1.32 | 0.94 | 68 | 47 | 3150 | 2246 | 0.35 |

Table 2: Number of parameters and corresponding sparsity and trained model test accuracy for MNIST (m1) architecture with various levels of scaling factor. The table reports the statistics of the experiment in Figure 1b.

| Scaling Factor | Total no. of Parameters | No. of 0-weights | Sparsity | Accuracy |
|---|---|---|---|---|
| 0.25 | $26,432$ | $5,946$ | 0.22 | 0.9063 |
| 0.50 | $54,912$ | $13,058$ | 0.24 | 0.9413 |
| 0.75 | $85,440$ | $18,703$ | 0.22 | 0.9544 |
| 1.00 | $118,016$ | $25,317$ | 0.21 | 0.9579 |
| 1.50 | $189,312$ | $43,428$ | 0.23 | 0.9618 |
| 2.00 | $268,800$ | $58,040$ | 0.22 | 0.9658 |
| 2.50 | $356,480$ | $83,451$ | 0.23 | 0.9676 |
| 3.00 | $452,352$ | $107,343$ | 0.24 | 0.9712 |

**Scale.** As discussed in Section B, to obtain high accuracy for binary and ternary models, the network needs to be scaled to achieve a higher model capacity. Figure 1b shows the impact of the scale of the network on accuracy of the model and its GC runtime for private inference. Although inference time increases linearly with the scaling factor, accuracy improves up to a certain extent with scaling (scaling factor of 1) and then becomes almost constant. This allows the designer to select a scaling factor that optimizes for both accuracy and performance.

Table 2 shows how the scaling factor affects the number of parameters in the network. As the scaling factor in TNNs increases, the accuracy also tends to increase as well. However, this levels off with diminishing returns after a certain limit. In addition to improved accuracy, the inference cost of the circuit also increases, as is evident from the growth of runtime with change in scaling factor. Note that the sparsity is 0.24 for a scaling factor of 3 for the ternary neural network, which means that the effective size of the model (hence its performance cost) remains comparable to a binary neural network (without any scaling), albeit with better accuracy. As a reference, the test accuracy of a BNN model with the same architecture is 0.9514.

**Comparison with prior work.** We present the outcome of basic SOTERIA (without the use of architecture search) on *fixed* model architectures which are used in the literature, thus only discussing the effect of sparsity of ternary neural networks on the tradeoff between accuracy and performance costs (shown in Table 3). We use three different architectures for each of the two datasets, which have been used in existing work. m1-3 are used with the MNIST dataset, while m4-6 are used with the CIFAR10 dataset. See Table 6 for the descriptions of model architectures. We use the same scaling factors for our networks as used by Riazi et al. (2019), which is the only other comparable work with quantized (binary) weights and inputs, for a fair comparison.

We observe that for MNIST datasets, the basic TNN SOTERIA models (m1-m3) provide improved runtime and communication performance *on average* than prior work with maximum drop in accuracy of only 0.0167 (for model m2). This shows that SOTERIA is useful in designing custom

models that provide optimal performance guarantees while retaining high prediction accuracy. For CIFAR10 datasets, we observe that for models used in prior work (m4 to m6), our basic TNN models exhibit a slightly higher drop in accuracy of $0.8$, but provide a computation and communication gain, on average, as compared to prior work. Overall, our results show that SOTERIA provides a flexible approach of training private models given the constraint on performance and accuracy of the model.

Table 3: Performance analysis of existing secure schemes for private neural network inference. We compare SOTERIA constructed on fixed model architectures with ternary parameters, as well as optimal architectures constructed by SOTERIA, with the prior work. We provide the descriptions of the model architectures in Table 6 . We use the same scaling factor for SOTERIA and XONN for fixed model architectures (1.75 for m1, 4.0 for m2, 2.0 for m3, 2.0 for m4, 3.0 for m5, 2.0 for m6), and use 3.0 for MNIST (SOTERIA) and CIFAR10 (SOTERIA) for the models constructed by our architecture search algorithm.

| Model | Secure Scheme | Runtime (s) | | | Communication (MB) | Test Accuracy | |
|---|---|---|---|---|---|---|---|
| | | Offline | Online | Total | | | |
| MNIST (m1) | SecureML Mohassel & Zhang (2017) | 4.7 | 0.18 | 4.88 | —[d] | 0.931 | |
| | MiniONN Liu et al. (2017) | 0.9 | 0.14 | 1.04 | 15.8 | 0.976 | |
| | EzPC Chandran et al. (2019) | — | —[c] | 0.7 | 76 | 0.976 | |
| | Gazelle Juvekar et al. (2018) | 0 | 0.03 | 0.03 | 0.5 | 0.976 | |
| | XONN Riazi et al. (2019) | — | —[c] | 0.13[b] | 4.29 | 0.976[a] | (0.9591) |
| | SOTERIA (TNN) | 0.04 | 0.03 | 0.07 | 3.72 | 0.9642 | |
| MNIST (m2) | DeepSecure Rouhani et al. (2018) | 7.69 | 1.98 | 9.67 | 791 | 0.9895 | |
| | MiniONN | 0.88 | 0.4 | 1.28 | 47.6 | 0.9895 | |
| | EzPC | — | —[c] | 0.6 | 70 | 0.990 | |
| | Gazelle | 0.15 | 0.05 | 0.20 | 8.0 | 0.990 | |
| | XONN | — | —[c] | 0.16[b] | 38.28 | 0.9864[a] | (0.9718) |
| | SOTERIA (TNN) | 0.08 | 0.06 | 0.14 | 30.68 | 0.9733 | |
| MNIST (m3) | MiniONN | 3.58 | 5.74 | 9.32 | 657.5 | 0.990 | |
| | EzPC | — | —[c] | 5.1 | 501 | 0.990 | |
| | Gazelle | 0.48 | 0.33 | 0.81 | 70 | 0.990 | |
| | XONN | — | —[c] | 0.15[b] | 32.13 | 0.990[a] | (0.9672) |
| | SOTERIA (TNN) | 0.08 | 0.07 | 0.15 | 26.04 | 0.9740 | |
| MNIST (SOTERIA) | SOTERIA ($\lambda = 0.6$) | 0.09 | 0.08 | 0.17 | 36.24 | 0.9883 | |
| | SOTERIA ($\lambda = 0$) | 0.016 | 0.018 | 0.034 | 95.18 | 0.9811 | |
| CIFAR10 (m4) | XONN | — | —[c] | 15.07[b] | 4980 | 0.80[a] | (0.7197) |
| | SOTERIA (TNN) | 8.56 | 6.14 | 14.70 | 936.1 | 0.7314 | |
| CIFAR10 (m5) | MiniONN | 472 | 72 | 544 | 9272 | 0.8161 | |
| | EzPC | — | —[c] | 265.6 | 40683 | 0.8161 | |
| | Gazelle | 9.34 | 3.56 | 12.9 | 1236 | 0.8161 | |
| | Delphi[e] | 45 | 1 | 46 | 200 | 0.85 | |
| | XONN | — | —[c] | 5.79[b] | 2599 | 0.8185[a] | (0.7266) |
| | SOTERIA (TNN) | 3.48 | 2.95 | 6.43 | 461.3 | 0.7252 | |
| CIFAR10 (m6) | XONN | — | —[c] | 16.09[b] | 5320 | 0.83[a] | (0.7341) |
| | SOTERIA (TNN) | 9.01 | 6.63 | 15.64 | 982.7 | 0.7396 | |
| CIFAR10 (SOTERIA) | SOTERIA ($\lambda = 0.6$) | 3.69 | 3.26 | 6.95 | 497.6 | 0.7384 | |
| | SOTERIA ($\lambda = 0$) | 4.01 | 3.53 | 7.54 | 561.2 | 0.7211 | |

[a]We could not reproduce the test accuracies for XONN. We report the results that we obtained on the same model architectures with the same setting in the respective paper in parenthesis. [b]XONN's runtime reported by the authors is measured on a high-performance Intel processor, which is faster than the one used by all other methods. [c]Breakdown of runtime cost into offline and online runtime is not reported by the authors. [d]Communication cost is not reported by the authors. [e]The runtime, communication cost and test accuracy are estimates from the graphs reported by the authors.

## 3.3 RESULTS WITH SOTERIA

We now discuss the details of our empirical analysis of SOTERIA. In particular, we will present the cost function we used in the architecture search algorithm, the effect of the performance regularization during the search, the effect of the model size on the tradeoff between accuracy and inference runtime, and compare SOTERIA with the prior work.

**Cost function for regularized architecture search.** Our algorithm searches for models that are not only accurate but are efficient with respect to the costs of using garbled circuits for computing the operations involved during inference time for a model. For this, we modify the score value generated by the generic architecture search algorithm Liu et al. (2019) gives to each operation (e.g., maxpool, or convolution with different dimensions) with a regularized penalty factor proportional to the performance cost of the operation. Table 4a presents the communication and runtime cost of

(a) Runtime and communication cost per operation.

| Operation | Runtime (ms) | Comm. (KB) | Penalty factor |
|---|---|---|---|
| CONV5 × 5 | 55.40 | 7942 | 1.00 |
| CONV3 × 3 | 23.10 | 3190 | 0.41 |
| MAXPOOL2 × 2 | 3.23 | 145 | 0.04 |
| IDENTITY | 0.00 | 0 | 0.00 |

(b) Number of parameters, sparsity and test accuracy.

| $\lambda$ | Total no. of Parameters | No. of 0-weights | Sparsity | Accuracy |
|---|---|---|---|---|
| 1.0 | $133,032$ | $41,212$ | 0.31 | 0.8892 |
| 0.8 | $729,768$ | $200,540$ | 0.27 | 0.9721 |
| 0.6 | $2,904,168$ | $1,080,217$ | 0.37 | 0.9883 |
| 0.4 | $2,904,168$ | $1,080,217$ | 0.37 | 0.9883 |
| 0.2 | $2,941,032$ | $895,034$ | 0.30 | 0.9887 |
| 0.0 | $11,466,600$ | $5,116,030$ | 0.45 | 0.9811 |

Table 4: (a) Runtime and communication cost of each operation based on their garbled circuit inference. We also calculate the performance penalty factor (which we use in our regularized architecture search algorithm) as the average of the relative costs for each unit w.r.t the most expensive operation. (b) Number of parameters and corresponding sparsity and test accuracy with different levels of $\lambda$ for SOTERIA architecture search over MNIST dataset with 1 cell architecture having 4 sequential operations. The setting is the same as the one illustrated in Figure 2(b). The scaling factor is 3. For larger $\lambda$, the algorithm penalizes large/complex models.

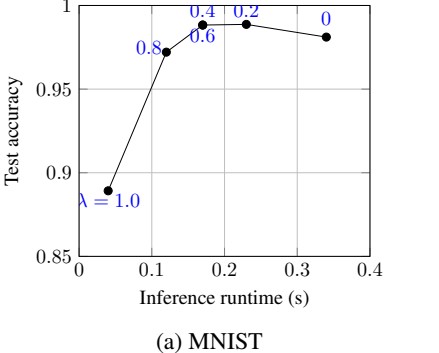
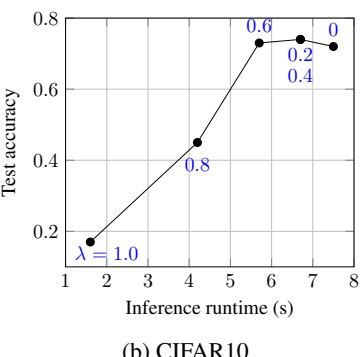

(a) MNIST       (b) CIFAR10

Figure 2: Inference runtime versus test accuracy of a garbled ternary model, constructed with SO-TERIA architecture search algorithm, for various values of circuit cost regularization $\lambda$. We obtain the architecture for a neural network with (a) 1 cell for MNIST dataset, and (b) 3 cells for CIFAR10 dataset. Scaling factor is 3.

each operation that we use in our algorithm. The penalty factor is computed as the average of the relative communication cost and relative runtime cost of each operation, with respect to the most costly operation (CONV5 × 5). We use this penalty factor in the experiments.

**Balancing accuracy and inference costs.** For the architecture search in SOTERIA, we balance accuracy and inference cost over GC protocol, using a regularization factor $\lambda$. With $\lambda = 1$ the importance of the penalty factor is maximum, and $\lambda = 0$ represents the case where we ignore the performance cost. We execute the search process with different values of $\lambda$. Figure 2 presents trade-off between test accuracy of the optimal architectures and their inference runtime. Table 4b provides the statistics on the number of model parameters and the model sparsity for different values of $\lambda$ for the MNIST dataset. As $\lambda$ increases, cheaper operations, that have fewer trainable parameters are chosen by the search process, which improves the inference runtime at the expense of the model accuracy. As we observe, $\lambda = 0.6$ provides a reasonable balance between both accuracy and the inference cost. The search algorithm identifies cheaper operations that collectively result in accuracy equivalent to using expensive operations. It is important to note that selecting $\lambda$ depends on how

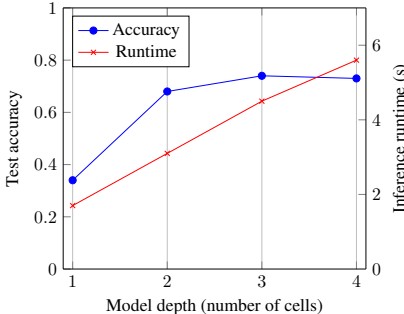

Figure 3: Impact of the model's depth (as the number of cells in the SOTERIA architecture search) on accuracy and inference runtime of garbled model on CIFAR10. Regularization term $\lambda$ is 0.6. Scaling factor is 3.

much cost or accuracy drop we can tolerate for a given setting. Thus, SOTERIA enables adapting private inference to the specific requirements and limitations of a system.

**Finding the optimal depth for the model (number of cells).** The number of cells that the final architecture has is a manually-set hyperparameter and defines the depth of the model architecture. We perform an experiment on the CIFAR10 dataset, using cells with 4 operations in sequence and $\lambda = 0.6$. We run the search process for 100 epochs, and train each resultant architecture for 200 epochs. Figure 3 presents the results of executing the search with different number of cells. It illustrates the model accuracies along with their inference runtime. We observe that for a single cell architecture, the accuracy levels are low, as the network could not process the features required to perform a generalizable classification. The test accuracy peaks for 3-cell architecture, suggesting that it has enough operations to process the required features of the inputs. In parallel, we can see that as the number of cells increases, the runtime also increases.

**Comparison with prior work.** Table 3 shows all the evaluated configurations in the prior work. It reports the results for SOTERIA trained models that are optimized for both accuracy and efficiency. For the MNIST dataset, we observe that our model with $\lambda = 0$ gives the best model as compared to prior work while balancing the runtime performance (0.034) and accuracy of 0.9811. Increasing $\lambda = 0.6$ increases the accuracy by a small value. Similarly, for CIFAR10 datasets, we observe that a SOTERIA-trained model with $\lambda = 0.6$ gives better accuracy than most of the prior work as compared to the numbers reported in brackets. Our evaluation on MNIST and CIFAR10 datasets confirm that SOTERIA is effective in training models that are customized to perform well for performance and accuracy by providing a set of options that trades-off these metrics in a user-configurable way.

## 4 RELATED WORK

Several prior work targets the problem techniques for secure machine learning, or build up on existing techniques by trying to optimize bottlenecks.

**Homomorphic Encryption.** In CryptoNets Dowlin et al. (2016)Xie et al. (2014), the authors modify the neural network operation by using square function as an activation and average pool instead of maxpool to reduce the non-linear functions to low degree polynomial to control the noise. Similar approaches of using homomorphic encryption on data and optimizing the machine learning operations to limit the noise have been explored extensively Bourse et al. (2017); Graepel et al. (2012); Bost et al. (2014). Hesamifard et al. Hesamifard et al. (2017b) explore using homomorphic encrypted data for training the neural networks. CryptoDL Hesamifard et al. (2017a) explores various activation functions with low polynomial degree that can work well with homomorphic encrypted data and proposed an activation using the derivative of ReLU function. However, using homomorphic encryption adds to an additional computational overhead and most of the non-linear activations cannot be effectively computed which results in a degradation of the deep learning systems.

**Secure Multiparty Computation.** Secure multiparty computation requires a very low computation overhead but requires extensive communication between the parties. It has been used for several

machine learning operations. DeepSecure Rouhani et al. (2018) only uses GC to compute all the operations in the neural network. They rely on pre-processing of the data by reducing the dimensions to improve the performance and is implemented on the TinyGarble Songhori et al. (2015) library. Chameleon Riazi et al. (2018) uses a combination of arithmetic sharing, garbled circuit and boolean sharing to compute the neural networks for secure inference. They rely on third party server to perform computation in the offline phase resulting in better performance than the previous work. XONN Riazi et al. (2019) leverages Binary Neural Networks with GC. Binarization dramatically reduces the inference latency for the network compared to other frameworks that utilize full-precision weights and inputs, as it converts matrix multiplications into simple XNOR-popcounts. They use TinyGarble library as well to implement the Boolean circuits for GC. Prio Corrigan-Gibbs & Boneh (2017) uses a secret sharing Shamir (1979) based protocol to compute aggregate statistics over private data from multiple sources. They deploy a secret-shared non-interactive zero-knowledge proof mechanism to verify whether data sent by clients is well-formed, and then decode summed encodings of clients' data to generate aggregate statistic. They extend the application of Prio to foundational machine learning techniques such as least squares regression.

**Hybrid Schemes.** A judicious combination of homomorphic encryption and multiparty computation protocol have shown to give some additional benefits in terms of runtime and communication costs. Gazelle Juvekar et al. (2018) uses lattice based Packed Additive homomorphic encryption to compute dot product and convolution but relies on garbled circuits for implementing non-linear operations like Maxpool and ReLU. They reduce the overall bandwidth by packing ciphertexts and re-encryption to refresh the noise budget. Delphi Mishra et al. (2020) builds upon this work and uses Architecture Search to select optimal replacement positions for expensive ReLU activation function with a quadratic approximation with minimal loss in accuracy. MiniONN Liu et al. (2017) pre-computes multiplication triplets using homomorphic encryption for GMW protocol followed by SPDZ Damgard et al. (2011; 2012) protocol. The multiplication triplets are exchanged securely using additive homomorphic encryption like Paillier or DGK. SecureML Mohassel & Zhang (2017) uses garbled circuits and additive homomorphic encryption to speed up some NN operations. However, the conversion costs between of homomorphic encryption and Yao's garbled circuits is expensive and the performance of homomorphic encryption scales poorly with increasing security parameter Demmler et al. (2015). Hence, we rely on only garbled circuit protocol to efficiently compute neural network operations during inference with low communication bandwidth, low computation complexity and low memory footprint using binary neural networks while maintaining the accuracy. Most of the previous work have relied heavily on optimizing the complex cryptographic operations to work well with the neural networks. We show that it is possible to optimize the neural network to get an efficient privacy preserving neural network architectures.

**Trusted Computing.** Some research uses trusted processors where they assume that the underlying hardware is trustworthy and outsource all the machine learning computations to the trusted hardware. Chiron Hunt et al. (2018a) is a training system for privacy-preserving machine learning as a service which conceals the training data from the operator. It uses Intel Software Guard Extensions (SGX) and runs the standard ML training in an enclave and confines it in a Ryoan sandbox Hunt et al. (2018b) to prevent it from leaking the training data outside the enclave. Ohrimenko et al. Ohrimenko et al. (2016) propose a solution for secure multiparty ML by using trusted Intel SGX-enabled processors and used oblivious protocols between client and server where the input and outputs are blinded. However, the memory of enclaves is limited and it is difficult to process memory and computationally intensive operations like matrix multiplication in the enclaves with paralellism. To address this, Slalom Tramèr & Boneh (2018) provides a methodology to outsource the matrix multiplication to a faster untrusted processor and verify the computation.

## 5 CONCLUSIONS

We introduce SOTERIA, a system that takes advantage of the power of neural architecture search algorithms in order to design model architectures which jointly optimize accuracy and efficiency for private inference. We construct optimal architectures that enables balancing accuracy and inference efficiency on garbled circuits. As opposed to the prior work that build cryptographic schemes around given fixed models, SOTERIA provides a flexible solution that can be adapted to the accuracy and performance requirements of any given system, and enables trading off between requirements.

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

# A    SELECTING THE CRYPTOGRAPHIC PRIMITIVE

In designing SOTERIA, we make several design choices with the goal of achieving efficiency. The most important among them is the selection of the underlying cryptographic primitive to ensure privacy of data. Several cryptographic primitives such as partially homomorphic encryption schemes (PHE) and fully homomorphic encryption schemes (FHE), Goldreich-Micali-Widgerson protocol (GMW), arithmetic secret sharing (SS), and Yao's garbled circuit (GC) have been proposed to enable two-party secure computation. Each of these primitives perform differently with respect to the factors such as efficiency, functionality, required resources and so on. Partially homomorphic encryption schemes allow either addition or multiplication operations but not both on encrypted data Paillier (1999); ElGamal (1985). In contrast, fully homomorphic encryption schemes schemes enable both addition and multiplication on encrypted data Gentry (2009b;a); van Dijk et al. (2010); Brakerski et al. (2011) but incur huge performance overhead. Secret sharing involves distributing the secret shares among non-trusting parties such that any operation can be computed on encrypted data without revealing the individual inputs of each party Beaver (1992). GMW Goldreich et al. (1987) and GC Yao (1982) allow designing boolean circuits and evaluating them between a client and a server. The differences between these schemes might make it difficult to decide which primitive is the best fit for designing a privacy-preserving system for a particular application. Therefore, we first outline the desirable properties specifically for private neural network inference and then compare these primitives with respect to these properties (see Table 5). We select a cryptographic scheme that satisfies all the following desired properties.

**Expressiveness.** This property ensures that the cryptographic primitive supports encrypted computation for a variety of operations. With the goal to enable private computation for neural networks, we examine the type of computations required in deep learning algorithms. Neural network algorithms are composed of linear and non-linear operations. Linear operations include computation required in the execution of fully-connected and convolution layers. Non-linear operations include activation functions such as Tanh, Sigmoid and ReLU. The research in deep learning is at its peak with a plethora of new models being proposed by the community to improve the accuracy of various tasks. Hence, we desire that the underlying primitive should be expressive with respect to any new operations used in the future as well. PHE schemes offer limited operations (either addition or multiplication) on encrypted data. This limits their usage in applications that demand expressive functionalities such as neural network algorithms. Alternative approaches such as FHE, SS, GMW and GC protocols allow arbitrary operations.

**Computation Efficiency.** Efficiency is one of the key factors while designing a client-server application such as a neural network inference service on the cloud. FHE techniques have shown to incur orders of magnitude overhead for computation of higher-degree polynmials or non-linear functions. Existing approaches using FHE schemes have restricted its use to compute only linear functions. However, most of the neural network architectures such as CNNs have each linear layer followed by a non-linear layer. To handle non-linear operations, previous solutions either approximate them to linear functions or switch to cryptographic primitives that support non-linearlity Dowlin et al. (2016); Mohassel & Zhang (2017); Juvekar et al. (2018); Mishra et al. (2020). Approximation of non-linear functions such as ReLU highly impacts the accuracy of the model. Switching between schemes introduces additional computation cost which is directly proportional to the network size. In comparison to FHE, research has shown that SS, GMW and GC schemes provide constructions with reasonable computation overhead for both linear and non-linear operations.

**Communication Overhead.** The communication costs incurred for private computation contribute to the decision of selecting our cryptographic primitive, as the network should not become a bottleneck in the execution of the private machine learning as a service. We expect the client and server to interact only once during the setup phase and at the end of the execution to receive the output. We aim to remain backward compatible to the existing cloud service setting where the client does not need to be online at all time between the request and response. In contradiction to this property, the GMW scheme requires communication rounds proportional to the depth of the circuit. To evaluate every layer with an AND gate, the client and server have to exchange secrets among them forcing the client to be online throughout the execution. Similarly, construction of non-linear bitwise functions with arithmetic secret shares require communication rounds logarithmic to the number of bits in the input. This makes the use of these schemes almost infeasible in the cloud setting that have a high-

|  | PHE | FHE | SS | GMW | GC |
|---|---|---|---|---|---|
| Expressiveness | × | ✓ | ✓ | ✓ | ✓ |
| Efficiency | ✓ | × | ✓ | ✓ | ✓ |
| Communication (One time setup) | ✓ | ✓ | × | × | ✓ |

Table 5: Properties of secure computation cryptographic primitives: Partially and fully homomorphic encryption schemes (PHE, FHE), Goldreich-Micali-Widgerson protocol (GMW), arithmetic secret sharing (SS), and Yao's garbled circuit (GC).

latency network. Unlike these primitives, Yao's garbled circuits combined with recent optimizations require an exchange of data only once at the beginning of the protocol.

We select Garbled Circuits as our underlying cryptographic primitive in SOTERIA as it satisfies all the desired properties for a designing private inference for cloud service applications.

## B  GARBLED CIRCUIT FOR EFFICIENT NEURAL-NETWORKS

We investigate the problem of performing private inference on neural networks. Let $W$ be the model parameters stored on the server, $x$ be the client's input, $y$ be the expected output and $f$ is the inference function to be computed. Given this, we want to compute $f(x; \theta) \to y$.

**Goals.** We aim for the following main goals:

- *Confidentiality:* The solution should preserve confidentiality of the model parameters $\theta$ from the users and that of $x$ and $y$ from the server. We assume an honest-but-curious threat model.

- *Accuracy:* The drop in accuracy of the privately computed inference function should be negligible as compared to the accuracy of the model on plaintext data.

- *Performance:* The private computation should demonstrate acceptable performance (runtime and communication) overhead.

**Garbled circuits.** GC protocol allows construction of any function as a boolean circuit with a one time setup cost of data exchange Yao (1986). In our setting, the client is the *garbler* and the server is the *evaluator*. In the setup phase, the client first transforms the function into a boolean circuit with two-input gates. The function (model architecture) and the circuit are known to both the parties, but its parameters and input are private. The client then garbles the circuit. This process involves creating a garbled computation table (GCT), which is an encrypted version of the truth table for the boolean circuit. The entries for this table are randomly permuted, so that the order does not leak information. The client then shares the garbled circuit and its encrypted inputs to the circuit (binary values representing $x$) with the server. In the next phase, the parties perform an oblivious transfer (OT) protocol Rabin (1981), so the server obtains the encryption of its inputs to the circuit (binary values representing $W$), without leaking information about its parameters to the client. Then, the server evaluates the circuit, and obtains the output value which is encrypted. The server transfers the output to the client which can match the encrypted values to their plaintext and obtain $f(x; \theta)$.

**Performance.** In GC, the communication and computation overhead is directly dependent on the number of AND, OR gates in the boolean circuit. Prior research has proposed several techniques that make it free for the GC to execute XOR, XNOR and NOT gates Kolesnikov & Schneider (2008). Given this prior work, the communication overhead of the GC protocol for a given circuit is proportional to its security parameter and the number of non-XOR gates in the circuit. The total runtime for evaluating a circuit is the sum of the time required during the **online** (evaluation) and **offline** (garbling and oblivious transfer) computation.

**Efficient neural networks.** In SOTERIA, we leverage the above-mentioned properties of GC to design an optimized neural network algorithm. Neural network algorithms are shown to be flexible with respect to their architectures i.e., multiple models with different configuration can achieve

a similar level of accuracy. We take advantage of this observation and propose designing neural network **architectures** that help optimize the performance of executing inference with garbled circuits. The number of gates in a circuit corresponding to a neural network depends on its activation functions and the size of its parameter vector.

Neural networks have shown to exhibit relatively high accuracy for various tasks even with low precision parameters. **Binary neural networks** Hubara et al. (2016) are designed with the lowest possible size for each parameter, i.e., one bit to represent $\{-1, +1\}$ values. Using BNNs naturally aligns with our selected cryptographic primitive because each wire in garbled circuits represents 1 bit value (representing $-1$ in the model with $0$ in the circuit). Binarizing the model parameters further allows us to heavily use the free XOR, XNOR and NOT gates in garbled circuits, thus minimizing the computation and communication overhead of private inference. This has recently been shown in the performance evaluation of garbled circuits on binary neural networks Riazi et al. (2019).

In neural networks, **linear functions** such as those used in the convolutional or fully connected layers form an important part of the network. These functions involve dot product vector multiplications. Instead of using multiplications, this can be computed very efficiently using XNOR-popcount: $\mathbf{x} \cdot \mathbf{w} = 2 \times \mathrm{bitcount}(\mathrm{xnor}(\mathbf{x}, \mathbf{w})) - N$, where $N = |\mathbf{x}|$. In binary neural networks, the output of activation functions is also binary. But, the output of XNOR-popcount is not a binary number, thus, according to BNN algorithms one would need to compare it with $0$; positive numbers will be converted to $1$ and negative numbers will be converted to $0$.

We can compute some **non-linear functions** such as maxpool very efficiently in BNNs. Maxpooling is a simple operation which returns the maximum value from a vector, which in the case of neural networks is usually a one-dimensional representation of a 2D max-pooling window. In binary neural networks, maxpool need to simply return $1$ if there is a $1$ in the vector. This is achieved by a logical OR-operation over the elements of the vector.

To achieve a learning capacity for binary neural networks similar to full-precision models, we would need to scale up the the number of model parameters. We can increase the number of kernels in a convolution layer and the number of nodes in a fully connected layer, by a given **scaling factor**. This technique has been used in the prior work Riazi et al. (2019), and enables learning more accurate models, however at the cost of increasing the number of computations in the network.

# C SPECIFICATIONS OF THE MODEL ARCHITECTURES

Table 6: Model architectures used in our experiments. Models m1-6 are used in the prior work on MNIST and CIFAR10 datasets, and we constructed the SOTERIA models using our regularized architecture search algorithm.

**MNIST (m1)**

| | Type | Kernels/Nodes |
|---|---|---|
| 1 | FC | 128 |
| 2 | FC | 128 |
| 3 | FC | 10 |

**MNIST (m2)**

| | Type | Kernels/Nodes |
|---|---|---|
| 1 | CONV $5 \times 5$ | 5 |
| 2 | FC | 100 |
| 3 | FC | 10 |

**MNIST (m3)**

| | Type | Kernels/Nodes |
|---|---|---|
| 1 | CONV $5 \times 5$ | 16 |
| 2 | MAXPOOL $2 \times 2$ | – |
| 3 | CONV $5 \times 5$ | 16 |
| 4 | MAXPOOL $2 \times 2$ | – |
| 5 | FC | 100 |
| 6 | FC | 10 |

**CIFAR10 (m4)**

| | Type | Kernels/Nodes |
|---|---|---|
| 1 | CONV $3 \times 3$ | 64 |
| 2 | CONV $3 \times 3$ | 64 |
| 3 | MAXPOOL $2 \times 2$ | – |
| 4 | CONV $3 \times 3$ | 64 |
| 5 | CONV $3 \times 3$ | 64 |
| 6 | MAXPOOL $2 \times 2$ | – |
| 7 | CONV $3 \times 3$ | 64 |
| 8 | CONV $1 \times 1$ | 64 |
| 9 | CONV $1 \times 1$ | 16 |
| 10 | FC | 10 |

**CIFAR10 (m5)**

| | Type | Kernels/Nodes |
|---|---|---|
| 1 | CONV $3 \times 3$ | 16 |
| 2 | CONV $3 \times 3$ | 16 |
| 3 | CONV $3 \times 3$ | 16 |
| 4 | MAXPOOL $2 \times 2$ | – |
| 5 | CONV $3 \times 3$ | 32 |
| 6 | CONV $3 \times 3$ | 32 |
| 7 | CONV $3 \times 3$ | 32 |
| 8 | MAXPOOL $2 \times 2$ | – |
| 9 | CONV $3 \times 3$ | 48 |
| 10 | CONV $3 \times 3$ | 48 |
| 11 | CONV $3 \times 3$ | 64 |
| 12 | MAXPOOL $2 \times 2$ | – |
| 13 | FC | 10 |

**CIFAR10 (m6)**

| | Type | Kernels/Nodes |
|---|---|---|
| 1 | CONV $3 \times 3$ | 16 |
| 2 | CONV $3 \times 3$ | 32 |
| 3 | CONV $3 \times 3$ | 32 |
| 4 | MAXPOOL $2 \times 2$ | – |
| 5 | CONV $3 \times 3$ | 48 |
| 6 | CONV $3 \times 3$ | 64 |
| 7 | CONV $3 \times 3$ | 80 |
| 8 | MAXPOOL $2 \times 2$ | – |
| 9 | CONV $3 \times 3$ | 96 |
| 10 | CONV $3 \times 3$ | 96 |
| 11 | CONV $3 \times 3$ | 128 |
| 12 | MAXPOOL $2 \times 2$ | – |
| 13 | FC | 10 |

**CIFAR10 (SOTERIA)**
$\lambda = 0$, No. of cells = 3, No. of operations per cell = 4

| | Type | Kernels/Nodes |
|---|---|---|
| 1 | CONV $5 \times 5$ | 16 |
| 2 | MAXPOOL $2 \times 2$ | – |
| 3 | CONV $5 \times 5$ | 16 |
| 4 | CONV $5 \times 5$ | 16 |
| 5 | CONV $5 \times 5$ | 32 |
| 6 | MAXPOOL $2 \times 2$ | – |
| 7 | CONV $5 \times 5$ | 32 |
| 8 | CONV $5 \times 5$ | 32 |
| 9 | CONV $5 \times 5$ | 64 |
| 10 | MAXPOOL $2 \times 2$ | – |
| 11 | CONV $5 \times 5$ | 64 |
| 12 | CONV $5 \times 5$ | 64 |
| 13 | FC | 10 |

**MNIST (SOTERIA)**
$\lambda = 0$, No. of cells = 1, No. of operations per cell = 4

| | Type | Kernels/Nodes |
|---|---|---|
| 1 | CONV $5 \times 5$ | 16 |
| 2 | CONV $5 \times 5$ | 16 |
| 3 | CONV $5 \times 5$ | 16 |
| 4 | CONV $5 \times 5$ | 16 |
| 5 | FC | 100 |
| 6 | FC | 10 |

**CIFAR10 (SOTERIA)**
$\lambda = 0.6$, No. of cells = 3, No. of operations per cell = 4

| | Type | Kernels/Nodes |
|---|---|---|
| 1 | CONV $3 \times 3$ | 16 |
| 2 | CONV $3 \times 3$ | 16 |
| 3 | CONV $3 \times 3$ | 16 |
| 4 | MAXPOOL $2 \times 2$ | – |
| 5 | CONV $3 \times 3$ | 32 |
| 6 | CONV $3 \times 3$ | 32 |
| 7 | CONV $3 \times 3$ | 32 |
| 8 | MAXPOOL $2 \times 2$ | – |
| 9 | CONV $3 \times 3$ | 64 |
| 10 | CONV $3 \times 3$ | 64 |
| 11 | CONV $3 \times 3$ | 64 |
| 12 | MAXPOOL $2 \times 2$ | – |
| 13 | FC | 10 |

**MNIST (SOTERIA)**
$\lambda = 0.6$, No. of cells = 1, No. of operations per cell = 4

| | Type | Kernels/Nodes |
|---|---|---|
| 1 | CONV $3 \times 3$ | 16 |
| 2 | CONV $3 \times 3$ | 16 |
| 3 | MAXPOOL $2 \times 2$ | – |
| 4 | CONV $5 \times 5$ | 16 |
| 5 | FC | 100 |
| 6 | FC | 10 |

# D   DETAILS FOR EXPERIMENTAL EVALUATION FOR PRIOR WORK

Table 7: Overview of the existing private inference methods, including the cryptographic schemes used, precision of neural networks supported, number of parties involved, evaluation setup configuration and availability of code.

| Prior Work | Cryptographic Scheme | Model Precision | Parties | Performance Evaluation Setup | | | Code |
|---|---|---|---|---|---|---|---|
| | | | | CPU | CPU Mark (Single Thread)[f] | Relative CPU Mark | |
| **MiniONN Liu et al. (2017)** | Additively HE, GC | Full | 2 | Server: Intel Core i5 4 × 3.30 GHz cores Client: Intel Core i5 4 × 3.20 GHz cores | 1,686-2,300 | 0.61-0.83 | Available[a] |
| **EzPC Chandran et al. (2019)** | GC, Additive SS | Full | 2 | Intel Xeon E5-2673 v3 2.40GHz | 1,723 | 0.62 | Available[b] |
| **DeepSecure Rouhani et al. (2018)** | GC | 16-bit fixed-point | 2 | Intel Core i7-2600 3.40GHz | 1,737 | 0.63 | – |
| **SecureML Mohassel & Zhang (2017)** | Linear HE, GC | Full | 2 | AWS c4.8xlarge (Intel Xeon E5-2666 v3 2.90 GHz) | 1,918 | 0.69 | – |
| **Gazelle Juvekar et al. (2018)** | Additively HE, GC | Full | 2 | AWS c4.xlarge (Intel Xeon E5-2666 v3 2.90GHz) | 1,918 | 0.69 | – |
| **Delphi Mishra et al. (2020)** | Additively HE, GC | Full | 2 | AWS c5.2xlarge (Intel Xeon 8000 series 3.0 GHz) | 2,082 | 0.75 | Available[c] |
| **SOTERIA** | GC | Ternary | 2 | AWS c5.2xlarge (Intel Xeon 8124M 3.0 GHz) | 2,082 | 0.75 | Available[d] |
| **Chameleon Riazi et al. (2018)** | GC, GMW, Additive SS | Full | 3[e] | Intel Core i7-4790 3.60GHz | 2,266 | 0.82 | – |
| **XONN Riazi et al. (2019)** | GC | Binary | 2 | Intel Core i7-7700k 4.5GHz | 2,777 | 1.0 | – |

[a] MiniONN: `https://github.com/SSGAalto/minionn`
[b] EzPC: `https://github.com/mpc-msri/EzPC`
[c] Delphi: `https://github.com/mc2-project/delphi`
[d] SOTERIA: `https://github.com/SoteriaAnonymous/Soteria`
[e] Chameleon only uses the third party in pre-processing stage.
[f] CPU mark: The configurations are listed with their the single-threaded CPU Mark scores as reported by `cpubenchmark.net/singleThread.html`. These single-thread benchmarks test processors on a variety of tasks, from floating point operations, string sorting and data compression (`https://www.cpubenchmark.net/cpu_test_info.html`) to provide an estimate of the capabilities of a processor. As microarchitectural optimizations vary from processor to processor, frequency alone cannot be used as a performance metric. The absolute scores and relative scores (compared to the highest scoring CPU in the table) for CPUs used in evaluation of related work are reported.

