# OpenReview forum: "Soteria: In search of efficient neural networks for private inference"
_ICLR.cc/2022/Conference — ICLR 2022 Submitted_

### Official Review · Reviewer_rwM8 · 2021-11-03

**Correctness:** 3
**Technical Novelty And Significance:** 2
**Empirical Novelty And Significance:** 2
**Recommendation:** 5
**Confidence:** 3

**Main Review:**

Pros:
1. The problem addressed in this paper is of practical importance for many real-world applications.
2. The challenges and the proposed solutions are well motivated.
3. The paper is also very well-written and has a nice flow.

Cons:

1.	The experiment section reads: “Although SOTERIA could leverage any of the available cryptographic primitives or their combination, we select garbled circuits as its main building block to address the confidentiality concern. Garbled circuits (GC) are known to be efficient and allow the generation of constant depth circuits even for non-linear functions. We show that neural network algorithms can be optimized to efficiently execute garbled circuits while achieving high accuracy guarantees.” It is not clear why faster GC+HE (homomorphic encryption) protocol than the pure-GC protocol that is proved by Gazelle [Chiraag Juvekar+, USENIX Security 2018] is not used.
2.	Multiple typos: For example, in the abstract, “multi-arty” should be “multi-party”.
3.	The main contribution of the paper is proposing a neural architecture search to automatically search for an optimal ternary neural network architecture. As such, the paper has a limited novelty from the ML perspective.
4.	The proposed method has shown improved performance over a few recent algorithms. The paper, however, fails to compare with CrypoNAS [Zahra Ghodsi+, NeurIPS 2020], which I think is the SOTA for cryptographic secure inference. It is better to compare more related works.


**Summary Of The Paper:**

This paper studies neural network designs for cryptographically secure inference. The paper uses the existing DARTS neural network architecture search algorithm to automatically a ternary neural network for secure inference based on garbled circuits.

**Summary Of The Review:**

--

---

### Official Review · Reviewer_TcBF · 2021-11-03

**Correctness:** 3
**Technical Novelty And Significance:** 3
**Empirical Novelty And Significance:** 3
**Recommendation:** 5
**Confidence:** 4

**Main Review:**

The main strength of the paper is its novelty, increasing the efficiency of cryptographic computations by neural architecture design is an interesting approach.

The main weakness is that the technical contribution is limited. It leverages the already existing techniques of neural architecture search and ternary networks to reduce the computation and communication overhead. There are also several points that could be clarified. For instance, it is not clear what is the cost of the architecture search process itself, in terms of the runtime. Is this included in the offline runtime in Table 3? Is the search repeated for different values of the regularization factor to find the best one (in terms of the trade-off between test accuracy and inference runtime), or is there a principled way for choosing the regularization factor to avoid this? If it is the former, this could be very costly, since the search should be repeated for various values assigned to the regularization parameter.


**Summary Of The Paper:**

The paper aims at developing an efficient neural network architecture with reduced computational cost under a cryptographic primitive where data on the client side and model on the server side are kept confidential. The motivation is that the existing works have focused on developing cryptographic techniques in a fixed network architecture and have not considered the neural network optimization perspective to enhance the efficiency of existing cryptographic computations. The paper then leverages the flexibility of the network structure while ensuring comparable accuracy and tries to find an architecture that provides a significant reduction in runtime while incurring negligible impact on prediction accuracy. Garbled circuits are chosen as the cryptographic primitive due to its compatibility with a wide range of computations, including non-linear functions. The paper then investigates how to reduce the private inference cost by incorporating two protocols: first one is neural architecture search for efficient private inference where a neural operation is penalized based on its computation and communication overhead, and the second one is to train models with ternary parameters which allows reducing the number of model parameters by introducing sparsity. Sparsity leads to a reduction in computation and communication overhead. However, to maintain a certain level of accuracy, the network needs to be scaled which incurs an increase in the runtime. Accordingly, a trade-off between efficiency and accuracy has been demonstrated with experiments on the CIFAR and MNIST datasets for varying regularization parameter and scaling factor.

**Summary Of The Review:**

Using neural architecture search to increase the efficiency of neural network architectures for private computing is a very interesting direction. Providing more insights on the cost of the search process and the optimization of the key parameters such as the regularization factor would help the reader better assess the benefits of neural architecture search in this context.

---

### Official Review · Reviewer_zXdE · 2021-11-08

**Correctness:** 2
**Technical Novelty And Significance:** 2
**Empirical Novelty And Significance:** 3
**Recommendation:** 3
**Confidence:** 2

**Main Review:**


The strength of the paper is the seemingly careful design that combines three elements (1) secure MPC protocol choice that reduces communication rounds, (2) restriction of weights to binary/ternary alphabet, and (2) compatabile neural architecture search to enable efficient private inference.

The main drawback of the paper is the lack of clarity in the exposition that makes the paper overall difficult to evaluate. The costs and drawbacks of the restricted architectures/weights compared to - say, a baseline with no privacy - is also not reported.
 Specifically:
- The neural architecture search is not explained rigorously via mathematics, only via words. This leaves the reader guessing as to what the authors are actually performing. Among many aspects where I am unclear, I am unable to understand how the cost functions change given that the architecture is now restricted to discrete set of weights and biases, as the overall training becomes a discrete optimization problem - so I expect the complexity of finding these neural networks and the weights to be enormous
- How is the factor lambda used in the cost function for the architecture search. The precise specification of the role of this factor is missing.
- It seems that the data set is utilized for both training the architecture, and then training the model. Intuitively, this must translate to more data being required as compared to training the model itself. The paper must describe this, and perhaps demonstrate any such effects through empirical results.
- The restriction to ternary weights and binary activation seems to be a significant drawback, and one expects that these restrictions must lead to weaker accuracy for some settings (data sets, or predictions). The paper misses a detailed discussion and experimental comparison in this regard.

In addition, it appears that the technical contribution builds on prior works, and so the novelty is relatively limited


**Summary Of The Paper:**

The paper develops a method for private neural-network-inference via utilization of Yao's garbled circuits (GC) protocol. In order to keep the computation complexity manageable - the main practical hurdle for  GC - the paper proposes utilization of a neural network architecture search, coupled with restricting weights to ternary alphabet and binary activation. The neural architecture search is conducted via a variation of the "DARTS" approach of Liu et. al (ICLR 2019), that accounts for model complexity in the cost function. The paper performs extensive experiments against comparable protocols in their evaluation for MNIST and CIFAR10, and show improvements along some factors.

**Summary Of The Review:**

Overall, even if the idea behind the paper is promising, given the unconventional approach to technical description (only words, not a single equation), and the associated lack of clarity and rigor, I recommend rejecting the paper in the current form.

---

### Decision · Program_Chairs · 2022-01-20

**Decision:**

Reject

**Comment:**

The authors propose a new way of addressing the ML as a service problem through using garbled circuits. As the reviewers point out the novelty is limited and comparison to existing work is not complete. The authors have also not responded to the reviews.